# A Higher Polygenic Risk Score Is Associated with a Higher Recurrence Rate of Atrial Fibrillation in Direct Current Cardioversion-Treated Patients

**DOI:** 10.3390/medicina57111263

**Published:** 2021-11-18

**Authors:** Simon Vogel, Irina Rudaka, Dmitrijs Rots, Jekaterīna Isakova, Oskars Kalējs, Kristīne Vīksne, Linda Gailīte

**Affiliations:** 1Scientific Laboratory of Molecular Genetics, Rīga Stradiņš University, 16 Dzirciema Str., LV-1007 Rīga, Latvia; irina.rudaka@rsu.lv (I.R.); dmitrijs.rots@rsu.lv (D.R.); jekaterina.isakova@rsu.lv (J.I.); kristine.viksne@rsu.lv (K.V.); linda.gailite@rsu.lv (L.G.); 2Latvian Cardiology Center, Pauls Stradiņš Clinical University Hospital, Pilsoņu iela 13, Zemgales priekšpilsēta, LV-1002 Rīga, Latvia; oskars.kalejs@rsu.lv

**Keywords:** atrial fibrillation, *PITX2*, *SOX5*, polygenic risk score, direct current cardioversion

## Abstract

*Background and Objectives*: Recurrence of atrial fibrillation (AF) within six months after sinus rhythm restoration with direct current cardioversion (DCC) is a significant treatment challenge. Currently, the factors influencing outcome are mostly unknown. Studies have found a link between genetics and the risk of AF and efficacy of rhythm control. The aim of this study was to examine the association between eight single-nucleotide variants (SNVs) and the risk of AF development and recurrence after DCC. *Materials and Methods:* Regarding the occurrence of AF, 259 AF cases and 108 controls were studied. Genotypes for the eight SNVs located in the genes *CAV1*, *MYH7*, *SOX5*, *KCNN3*, *ZFHX3*, *KCNJ5* and *PITX2* were determined using high-resolution melting analysis and confirmed with Sanger sequencing. Six months after DCC, a telephone interview was conducted to determine whether AF had recurred. A polygenic risk score (PRS) was calculated as the unweighted sum of risk alleles. Multivariate regression analyses were performed to assess SNV and PRS association with AF occurrence and recurrence after DCC. *Results:* The risk allele of rs2200733 (*PITX2*) was significantly associated with the development of AF (*p* = 0.012, OR = 2.31, 95% CI = 1.206–4.423). AF recurred in 60% of patients and the allele generally associated with a decreased risk of AF of rs11047543 (*SOX5*) was associated with a greater risk of AF recurrence (*p* = 0.014, OR = 0.223, 95% CI = 0.067–0.738). A PRS of greater than 7 was significantly associated (*p* = 0.008) with a higher likelihood of developing AF after DCC (OR = 4.174, 95% CI = 1.454–11.980). *Conclusions*: A higher PRS is associated with increased odds of AF recurrence after treatment with DCC. *PITX2* (rs2200733) is significantly associated with an increased risk of AF. The protective allele of rs11047543 (*SOX5*) is associated with a greater risk of AF recurrence. Further studies are needed to predict the success of rhythm control and guide patient selection towards the most efficacious treatment.

## 1. Introduction

The most common arrhythmia is atrial fibrillation (AF), a condition that, if left untreated, can potentially lead to fatal complications [1,2].

Genetic studies have identified common single-nucleotide variants (SNVs) that are associated with AF [3,4,5,6,7,8,9,10]. Furthermore, effort has been directed towards developing polygenic risk scores (PRS) for certain therapies, with the aim of predicting the potential efficacy based on the genotypes of patients [11,12,13].

The main pillars of AF treatment are rate and rhythm control as well as anticoagulation [14]. Recent work has shown a higher AF recurrence rate after treatment with catheter ablation in patients with a higher PRS [13]. However, to date, there are no data on whether there is an association between AF recurrence after treatment with direct current cardioversion (DCC) and a higher PRS. It is estimated that 47% of patients that have an episode of AF revert to arrhythmia within six months after sinus rhythm restoration [15]. Consequently, this study aimed to examine the association between eight common SNVs previously reported to be associated with AF and the actual risk of AF development and recurrence after DCC treatment.

## 2. Materials and Methods

### 2.1. Study Group

The study was performed according to the guidelines of the Declaration of Helsinki and was approved by the Latvian Central Medical Ethics Committee (No. 1/16-05-09). Prior to enrolment in the study, participants signed an informed consent form.

The case group was comprised of 259 patients with persistent and long-standing persistent symptomatic AF, who were treated with DCC at the Latvian Cardiology Center of Pauls Stradiņš Clinical University Hospital. Additionally, 108 control individuals were recruited from general practitioner offices as well as from the Internal Disease Department at Madona Hospital in Latvia. The majority of patients were pretreated with anti-arrhythmic drugs (AAD) before undergoing DCC—20% received class Ic AAD (Ethacizine or Propafenone) and 4% class II (sotalol) and 55% class III (Amiodarone). In the case of successful sinus rhythm restoration, patients remained on AAD for up to three months. The efficacy of cardioversion was determined by the presence of symptomatic AF after six months via an interview. Data were available for 97 patients.

In 50 of these 97 patients, transthoracic echocardiography data were available (Appendix A). Investigations were performed in outpatient clinics and were carried out according to a standard protocol approved by the Latvian Society of Cardiology. Two echocardiographic parameters (left atrial volume index (LAVI, mL/m^2^) and left ventricular ejection fraction (EF, %)) were used in our analyses.

### 2.2. Genetic Analysis

For all participants, DNA was extracted from peripheral blood samples using the commercially available innuPREP Blood DNA Mini Kit (Analytik Jena AG, Jena, Germany) [9].

Eight SNVs previously reported to be associated with AF in genome-wide association studies (GWAS) and with a minor allele frequency (MAF) of >5% in Europeans (gnomAD v.2.0.1) were selected for genotyping (Table 1). The selected SNVs were not in strong linkage disequilibrium.

Genotyping was carried out by real-time PCR and high-resolution melting analysis using Rotor-Gene Q (QIAGEN N.V., Venlo, The Netherlands), as described previously [9]. Primers are available upon request. To confirm the genotyping results, 8–16 samples with different genotypes were randomly selected for Sanger sequencing using the BigDye Terminator Kit v3.1 (Thermo Fisher Scientific, Waltham, MA, USA).

### 2.3. Statistical Analysis

Data selection, checking for multicollinearity using the variance inflation factor and heatmaps, and parts of the visualization were completed using Python programming language (version 3.8) on Spyder (version 5.0.3), as well as Microsoft Office Excel software. For the descriptive statistics and logistic regression analyses, SPSS software (version 27.0) was used. To determine differences between the case and control groups, a χ^2^ test was used for the categorical variables, whereas for the continuous variables, a logistic regression was performed.

Two multivariate logistic regression analyses were performed on the data of the case (*n* = 259) and control (*n* = 108) groups: first, to assess the association between the presence of a risk allele and the development of AF (Figure 1), and second, to evaluate the association between a higher PRS and the development of AF (Appendix A). Independent variables associated with AF that had fewer than five instances in either the case or control group were excluded from the study, with the significance being defined as *p* < 0.05. Additionally, only variables known to be associated with an increased risk of developing AF were included in the models [17,18,19,20,21,22,23]. We endeavored to adhere to the rule of 10 independent variables per case to comply with best practice [24].

To determine the relationship between the presence of risk alleles as well as the PRS and the recurrence of AF after DCC treatment, two multivariate regression analyses were performed on a group of 97 AF patients with known DCC outcomes: first, the risk alleles were examined individually (Figure 2), and second, the association between a higher PRS and the recurrence of AF was evaluated (Figure 3). Regarding the variables included in these analyses, the variables ‘duration of AF’ and ‘age of onset’ were included as independent variables. Furthermore, to reduce the number of independent variables, a CHA2DS2–VASc score of >2 for women and >1 for men was used to combine risk factors for stroke in the setting of AF. Specifically, the CHA2DS2–VASc score was used to combine the variables ‘sex’, ‘age’, ‘congestive heart failure’, ‘previous stroke’ and ‘diabetes’. Another variable was created to combine other comorbidities associated with AF, such as ‘pulmonary arterial hypertension’, ‘dyslipidemia’, ‘chronic respiratory disease’ and ‘coronary heart disease’.

Finally, a multiple logistic regression analysis was performed on a subgroup of 50 patients to assess the effects of common echocardiographic parameters on AF recurrence after DCC, where all comorbidities were combined into a single variable termed ‘comorbidities’.

The PRS was calculated as the unweighted sum of risk alleles based on all the SNVs. In line with other studies [25,26,27,28], if there was an association with a risk allele at *p* < 0.1, the risk allele was flipped so that the major allele was considered to be the risk allele to reflect this trend within our patient population. We chose *p* < 0.1 as this was the value adopted as the threshold for inclusion of SNVs in the PRS by Choe et al. in their study of PRS and the efficacy of catheter ablation in the setting of AF [13]. A score of greater than 7 was chosen, as the highest score a subject had was 9 out of a theoretical maximum score of 16.

## 3. Results

Description of the case (*n* = 259) and control (*n* = 108) groups is shown in Table 2. Several variables differed significantly between the two groups, e.g., sex, age and body mass index (BMI), with the case group having a greater proportion of men, being older and having a higher BMI. In addition, the case group had a higher share of individuals with congestive heart failure, coronary heart disease and dyslipidemia. The control group included relatively more individuals with diabetes. Regarding the variables included in the multivariate regression analysis assessing AF recurrence after DCC (*n* = 97), only sex differed significantly (Table 3). When considering the subgroup of patients with transthoracic echocardiography data (*n* = 50), none of the variables included in the multivariate regression analysis differed significantly (Table 4). 

### 3.1. Multiple Regression Analysis of the Case and Control Groups Regarding the Risk of Developing AF for Each SNV and for a PRS of >7

In the multivariate logistic regression analysis that included factors associated with the development of AF (Appendix A), only one SNV—rs2200733 (*PITX2*)—was significantly associated (*p* = 0.012) with the development of AF (OR = 2.31, 95% CI = 1.206–4.423) (Figure 1). The variables age, sex (male), BMI, congestive heart failure, coronary heart disease, diabetes (type 1 and 2) and dyslipidemia were significantly associated with the development of AF (*p* < 0.05) (Appendix A). A PRS of >7 was not associated with a higher likelihood of developing AF (Appendix A).

### 3.2. Multiple Regression Analysis of AF Patients with Known DCC Outcomes for the Risk of AF Recurrence for Each SNV and for a PRS of >7

Next, a multivariate logistic regression analysis that included factors associated with the recurrence of AF was conducted (Appendix A). Only one SNV—rs11047543 (*SOX5*)—was significantly associated (*p* = 0.014) with the recurrence of AF (OR = 0.223, 95% CI = 0.067–0.738) (Figure 2). Additionally, rs2106261 (*ZFHX3*) and rs6838973 (*PITX2*) were associated with the recurrence of AF at a level of *p* < 0.1. Of the other independent variables, only the duration of AF was significantly associated (*p* = 0.014) with recurrence (OR = 1.015, 95% CI = 1.003–1.028) (Appendix A).

The SNVs rs2106261 (*ZFHX3*) and rs11047543 (*SOX5*) were flipped as there was a negative association between the presence of the risk allele and the recurrence of AF at a *p* value of <0.1. A PRS of >7 was significantly associated (*p* = 0.008) with a higher likelihood of the recurrence of AF after DCC (OR = 4.174, 95% CI = 1.454–11.980), as was the duration of AF (*p* = 0.026, OR = 1.014, 95% CI = 1.002–1.026) (Figure 3, Appendix A).

### 3.3. Multiple Regression Analysis of AF Patients with Known DCC Outcomes and Transthoracic Echocardiography Data (n = 50) for the Risk of AF Recurrence for a PRS of >7

Neither the EF (OR = 1.011, 95% CI = 0.930–1.099) nor the LAVI (OR = 1.105, 95% CI = 0.990–1.233) were significantly associated with an increased risk of AF recurrence. However, a PRS of >7 (*p* = 0.014, OR = 38.766, 95% CI = 2.085–720.952) and male sex (*p* = 0.013, OR = 18.569, 95% CI = 1.839–187.492) were significantly associated with an increased risk of AF recurrence (Appendix A).

## 4. Discussion

### 4.1. Discussion of Results

The aim of this study was to establish whether there is indeed an association between common genetic variants previously reported to be linked to AF and the actual risk of AF development and recurrence after DCC treatment. When assessing the occurrence of AF in our study population by examining the case and control groups, we found that the T risk allele rs2200733 (*PITX2*) was significantly associated with an increased risk of occurrence. Our finding is in line with the majority of similar studies [7,9,10,11,12,13,29,30,31,32,33,34]. The seven other SNVs described in GWAS were not found to be associated with a greater risk of AF development. This result may be due to rs2200733 (*PITX2*) potentially playing a greater role in the pathophysiology of AF, there being an association only in this patient population or it being an anomaly. For example, Huang and Darbar found that the SNVs associated with AF occurrence were different depending on the population studied in a GWAS [35]. The SNVs chosen for evaluation in this study reflected the SNVs commonly associated with AF in GWAS examining European populations. However, these GWAS had predominantly Western European individuals, whereas the population group of our study was predominantly recruited from Eastern Europe [36]. While we found an association only with the highest ranking SNV, it is possible that the other SNVs’ effects were too small to be detected in our cohort.

Interestingly, while rs2200733 (*PITX2*) was the only SNV in our study associated with the risk of developing AF, it was not significantly associated with an increased risk of AF recurrence in a subgroup of cases that had DCC outcome data. However, as the protective A allele of the rs11047543 (*SOX5*) SNV and the protective T allele of the rs2106261 (*ZFHX3*) SNV reached the risk of recurrence threshold of *p* < 0.1 and were consequently flipped, it was subsequently found that a higher PRS was indeed associated with a higher chance of AF recurring after DCC treatment (OR = 4.174, 95% CI = 1.454–11.980). This result was even more evident when a further subgroup of 50 cases for which transthoracic echocardiography data were available underwent multiple logistic regression analysis (OR = 38.766, 95% CI = 2.085–720.952). This raises the question as to whether one should flip a risk allele for the PRS in order to match a general trend in the population, perhaps at the expense of the generalizability of the results for other populations. An argument can be made that the adjustment of a risk allele for PRS scoring would allow for a tailored approach for each population. Indeed, several studies have already adopted this approach [25,26,27,28,37]. 

In accordance with our aforementioned finding that a higher PRS is associated with a higher risk of AF recurrence after DCC treatment, Choe et al. also found a higher PRS using five SNVs to be associated with a higher risk of AF recurrence after catheter ablation treatment [13]. Furthermore, O’Sullivan et al., studying 530,933 SNVs and Pulit et al. studying 934 SNVs, found a link between a higher PRS and a higher risk of stroke in AF patients. Additionally, Kertai et al., investigating 2,746 SNVs, detected an association between a higher PRS and a higher risk of developing AF post cardiac surgery [38,39,40]. Similar to this study, multiple studies have observed the trend of significant association of AF recurrence only with a higher PRS and generally not with the SNVs themselves. One likely reason for this is that the SNVs associated with AF are frequently located outside of coding regions and thus not directly involved in the pathogenesis of AF [41,42]. Indeed, all eight SNVs investigated here are located in non-coding regions. The current consensus is that a PRS should include the maximum number of SNVs possible as the pathogenesis of AF is polygenic and not yet fully understood despite being an area of active research [43]. As such, recent studies have utilized whole-genome sequencing in conjunction with GWAS to permit the calculation of PRS using a far greater number of SNVs [37,38,39,40,44,45,46,47,48]. This has been made possible by the increasing availability of whole-genome sequencing [37,41,42,49].

### 4.2. Limitations

Thus, one limitation of the present study is the number of SNVs comprising the PRS. In Choe et al.’s study that reported an association between a higher PRS and an increased risk of AF recurrence after catheter ablation treatment, 20 SNVs were initially considered for the calculation of PRS; however, only five SNVs that reached a threshold of *p* < 0.1 were ultimately included in the PRS [13]. Here, all eight SNVs were automatically included in the PRS. As it is now generally accepted that a PRS is best calculated using the greatest number of SNVs, future research investigating the recurrence of AF after DCC treatment, as well as after catheter ablation treatment, should include a much greater number of SNVs whose genotypes are determined by whole-genome sequencing [37].

Another limitation of the study is the sample size and a possible selection bias. There was a difference between the case and control groups with regards various variables (sex, age, BMI, congestive heart failure, coronary heart disease and diabetes) to occurrence, which hints at a selection bias. Since the results of the regression analysis only confirmed the result that rs2200733 (*PITX2*) was associated with a higher risk for AF occurrence, this result simply echoes previous studies [7,9,10,11,12,13,29,30,31,32,33,34]. Additionally, we found that as the number of cases included in the multiple regression analysis decreased, the odds ratio and confidence interval increased from OR = 4.174 (95% CI = 1.454–11.980) for the 97 patients treated with DCC for whom follow-up data were available to OR = 38.766 (95% CI = 2.085–720.952) for the 50 patients for whom transthoracic echocardiography data were also available. This raises the question as to whether the results would persist if the data for AF recurrence and transthoracic echocardiography were available for all 259 AF patients that underwent DCC rhythm control treatment.

Finally, a limitation was the fact that persistent symptomatic AF recurrence was only determined by telephone interview as this study was conducted during the COVID19 pandemic [50,51,52].

### 4.3. Broad View

Nevertheless, this study shows that a higher PRS using the eight selected SNVs is associated with a decreased efficacy of DCC to treat AF patients. As genetic testing is becoming more readily available and cheaper, this could therefore be a first step towards larger studies that will help determine the best treatment options for a patient with AF based on their PRS [53].

## 5. Conclusions

We conclude that a higher PRS of SNVs rs3807989 (*CAV1*), rs11047543 (*SOX5*), rs28631169 (*MYH7*), rs2106261 (*ZFHX3*), rs13376333 (*KCNN3*), rs75190942 (*KCNJ5*), rs2200733 (*PITX2*) and rs6838973 (*PITX2*) is likely associated with increased odds of AF recurrence after successful sinus rhythm restoration with DCC. Additionally, it can be concluded that of the eight SNVs analyzed, only rs2200733 (*PITX2*) was significantly associated with an increased risk of developing AF. Further studies are needed to predict the success of rhythm control and guide patient selection towards the most efficacious treatment, especially since, to the best of our knowledge, this is the first study to investigate the link between a PRS and the recurrence of AF after DCC treatment. 

## Figures and Tables

**Figure 1 medicina-57-01263-f001:**
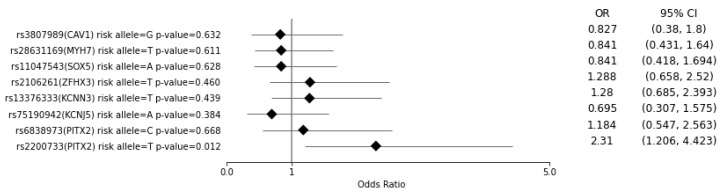
AF occurrence risk per single tested variant. The odds ratio (OR) and 95% confidence interval (CI) are from the multivariate logistic regression analysis (*n* = 367) (Appendix A).

**Figure 2 medicina-57-01263-f002:**
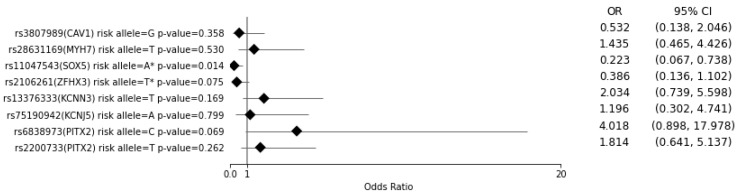
AF recurrence risk after DCC treatment per single tested variant. The odds ratio (OR) and 95% confidence interval (CI) are from the multivariate logistic regression analysis (*n* = 97) (Appendix A). * Alleles with *p* < 0.1 and OR < 1 consequently flipped for PRS.

**Figure 3 medicina-57-01263-f003:**
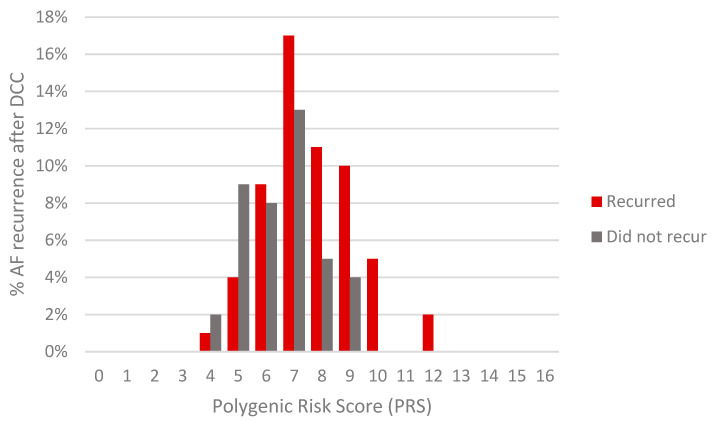
AF recurrence risk based on PRS. The PRS was calculated as the total unweighted number of risk alleles that DCC-treated patients with and without AF recurrence had (*n* = 58 patients and *n* = 39 patients, respectively).

**Table 1 medicina-57-01263-t001:** The eight SNVs selected for genotyping and their corresponding genome-wide association studies reporting an association with AF.

Gene	Locus	SNV	Risk Allele	MAF	Location	GWAS Reporting an AF Link
*CAV1*	7q31	rs3807989	G	0.61	Intron	Ellinor et al., 2012 [3]
*SOX5*	12p12	rs11047543	A	0.10	Upstream	Pfeufer et al., 2010 [4]
*MYH7*	14q11	rs28631169	T	0.12	Intron	Roselli et al., 2018 [5]
*ZFHX3*	16q22	rs2106261	T	0.24	Intron	Ellinor et al., 2012 [3]
*KCNN3*	1q21	rs13376333	T	0.27	Intron	Ellinor et al., 2010 [16]
*KCNJ5*	11q24	rs75190942	A	0.09	Downstream	Christophersen et al., 2017 [7]
*PITX2*	4q25	rs2200733	T	0.15	Upstream	Gudbjartsson et al., 2007 [8]
*PITX2*	4q25	rs6838973	C	0.43	Upstream	Other studies: Rudaka et al., 2020, Kiliszek et al., 2011 [9,10]

**Table 2 medicina-57-01263-t002:** Description of the case and control groups.

Variable	Cases (*n* = 259)	Controls (*n* = 108)	*p* Value ^1^	OR ^2^	95% Confidence Interval
Sex Male, *n* (%) Female, *n* (%)	99 (38.2) 160 (61.8)	39 (36.1) 69 (63.9)	**<0.001**	0.350	0.219–0.557
Age ^3^, years	64.5 ± 9.77	61.4 ± 9.92	**0.007**	1.032	1.009–1.056
Body Mass Index ^3^, kg/m^2^	31.3 ± 5.41	29.5 ± 5.44	**0.005**	1.066	1.019–1.116
Pulmonary Arterial Hypertension, *n* (%)	201 (77.6)	80 (74.1)	0.467	1.213	0.721–2.040
Congestive Heart Failure, *n* (%)	161 (62.2)	14 (13.0)	**<0.001**	11.031	5.963–20.404
Coronary Heart Disease, *n* (%)	50 (19.3)	3 (2.8)	**<0.001**	8.373	2.551–27.479
Stroke, *n* (%)	11 (4.2)	8 (7.4)	0.213	0.554	0.217–1.419
Diabetes (Type 1 and 2), *n* (%)	24 (9.3)	20 (18.5)	**0.013**	0.449	0.236–0.854
Dyslipidemia, *n* (%)	86 (33.2)	64 (59.3)	**<0.001**	0.342	0.215–0.543
Chronic Respiratory Disorders, *n* (%)	17 (6.6)	5 (4.6)	0.477	1.447	0.520–4.027

^1^ Statistically significant *p* values are shown in bold. ^2^ χ^2^ test was used for the categorical variables and a simple logistic regression was used for the continuous variables. ^3^ Data represent mean ± standard deviation.

**Table 3 medicina-57-01263-t003:** Characterization of patients from DCC follow-up for whom AF recurred vs. did not recur.

Variable ^1^	AF Recurred (*n* = 58)	AF Did Not Recur (*n* = 39)	*p* Value ^2^	OR ^3^	95% Confidence Interval
Sex Male, *n* (%) Female, *n* (%)	35 (60.3) 23 (39.7)	31 (79.5) 8 (20.5)	**0.047**	2.546	0.996–6.509
Age ^4^, years	61.7 ± 10.1	62.1 ± 10.4	0.882	0.997	0.958–1.038
Body Mass Index ^4^, kg/m^2^	31.5 ± 5.66	31.6 ± 5.77	0.954	0.998	0.929–1.072
Duration Since Initial Diagnosis ^4^, months	58.6 ± 95.8	27.9 ± 43.2	0.059	1.010	1.000–1.021
Age At Initial Diagnosis ^4^, years	56.9 ± 12.6	59.8 ± 10.4	0.247	0.979	0.945–1.015
Pulmonary Arterial Hypertension, *n* (%)	43 (74.1)	27 (69.2)	0.597	1.274	0.519–3.130
Congestive Heart Failure, *n* (%)	38 (65.5)	26 (66.7)	0.768	0.877	0.367–2.098
Coronary Heart Disease, *n* (%)	7 (12.1)	7 (17.9)	0.419	0.627	0.201–1.956
Stroke, *n* (%)	3 (5.2)	2 (5.1)	0.992	1.009	0.161–6.335
Diabetes (Type 1 and 2), *n* (%)	4 (6.9)	7 (17.9)	0.339	0.092	0.092–1.247
Dyslipidemia, *n* (%)	21 (36.2)	16 (41.0)	0.632	0.816	0.355–1.877
Chronic Respiratory Disorders, *n* (%)	3 (5.2)	4 (10.3)	0.343	0.477	0.101–2.262
CHA2DS2–VASc, *n* (%)	50 (86.2)	37 (94.9)	0.169	0.338	0.068–1.685
Non-CHA2DS2–VASc Comorbidities, *n* (%)	47 (81.0)	36 (92.3)	0.121	0.356	0.092–1.371

^1^ Variables used in the subsequent multivariate logistic regression analysis are presented here. ^2^ Statistically significant *p* value is shown in bold. ^3^ χ^2^ test was used for the categorical variables and a simple logistic regression was used for the continuous variables. ^4^ Data represent mean ± standard deviation.

**Table 4 medicina-57-01263-t004:** Characterization of patients from DCC follow-up for whom AF recurred vs. did not recur with transthoracic echocardiography data.

Variable ^1^	AF Recurred (*n* = 35)	AF Did Not Recur (*n* = 15)	*p* Value	OR ^2^	95% Confidence Interval
Sex Male, *n* (%) Female, *n* (%)	14 (40.0) 21 (60.0)	2 (13.3) 13 (86.7)	0.064	4.333	0.845–22.230
Comorbidities, *n* (%)	30 (85.7)	15 (100)	0.123	1.500	1.220–1.844
Body Mass Index ^3^, kg/m^2^	32.5 ± 5.95	33.8 ± 6.35	0.484	0.965	0.872–1.067
Duration Since Initial Diagnosis ^3^, months	58.3 ± 112	31.6 ± 60.0	0.425	1.005	0.993–1.018
Age At Initial Diagnosis ^3^, years	56.1 ± 12.9	61.2 ± 12.9	0.206	0.967	0.919–1.019
Left Atrial Volume Index ^3^, mL/m^2^	41.6 ± 11.8	40.8 ± 8.44	0.818	1.007	0.950–1.066
Ejection Fraction ^3^, %	54.1 ± 11.7	53.9 ± 8.45	0.956	1.002	0.946–1.060

^1^ Variables used in the subsequent multivariate logistic regression analysis are presented here. ^2^ χ^2^ test was used for the categorical variables and a simple logistic regression was used for the continuous variables. ^3^ Data represent mean ± standard deviation.

## Data Availability

Data are available on request.

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
