# Peer review of "A Higher Polygenic Risk Score Is Associated with a Higher Recurrence Rate of Atrial Fibrillation in Direct Current Cardioversion-Treated Patients"

_medicina, 2021, doi:10.3390/medicina57111263_

Round 1

Reviewer 1 Report

the paper is of good quality and original in its contents. the English language could improve

Author Response

Thank you for the comment - English were corrected by native speaker using Clark Scientific editing http://www.clarkscientificediting.com/ service. If there is needed approval we can add it.

Reviewer 2 Report

Unfortunately the study suffers from many serious methodological errors, which cannot be fixed. To name the most important ones:

  • study group and AF group differ significantly, even in terms of baseline characteristics such as age and sex 
  • follow-up is based on the presence of symptomatic AF after six months via an interview only. It is unacceptable, there might have been asymptomatic AF runs etc, which makes all of the calculations useless. There should be also some form of objective assessment.
  • data on echo were available only for 97 patients (37%). It is the most basic study to use in AF for diagnostic and risk stratification purposes. Even less echo were available during follow-up.
  • the numbers are overall small for nowadays SNPs study 
  • why only patients with persistent and long-standing AF were included and not with paroxysmal AF - it is not explained
  • exactly which factors were included in the PRF? It is hard to tell.
  • the mechanisms through which analyzed SNPs could affect AF onset or recurrence are not presented or proposed
  • abstract: line 21 - recurrence of AF is not an efficacy of DCC
  • abstract: line 31 - this sentence is illogical. 

Reviewer 3 Report

This study analyzed the relationship between AF recurrence after DC cardioversion and polygenic risk score using well-known SNVs, which are related to AF. Thank you for allowing me to review this interesting study.

Major

1. Table 3

Because the 95% CI of the variable "Sex" contains 1, its p-value cannot be less than 0.05. However, the p-value of the variable "Sex" is 0.047, which is less than 0.05. This is an error, please correct it.

The difference in the variable “duration since the initial diagnosis, months” between the AF recurred group and the AF Did Not recur group was very large, and it's 95% CI was 1.000-1.021, which does not include 1. Therefore, the p-value of this variable cannot be 0.590.

Check all statistics in Table 3 and correct any errors.

2. Please provide information on antiarrhythmic drugs used during and after DCC and adjust it in regression analysis.

A major limitation of this study is the lack of information on the antiarrhythmic drugs used during and after DCC. The type and dose of the antiarrhythmic drug make a significant effect on the risk of AF recurrence after DCC. If not all patients did DCC without antiarrhythmic drugs, information on the type and amount of antiarrhythmic drugs used should be provided and this information should also be adjusted in regression analysis.

Minor

Please change “chronic heart failure” to “congestive heart failure”

Round 2

Reviewer 2 Report

None

Author Response

Dear reviewer,

Thank you so much for your comments!

We have read through the entire paper again, in order to look for any other grammar/spelling as well as phrasing errors that our editor may have missed and changed a few things to hopefully address your concerns.

If you have found any errors or things that are unclear, do let us know!

Kind regards,

Simon Vogel

Reviewer 3 Report

The authors have responded well to my comments. thank you.

Author Response

Dear reviewer,

Thank you so much for your comments! We are glad we could address your concerns.

If you have any other suggestions regarding possible changes, do let us know!

Kind regards,

Simon Vogel